

# Comparison of land-atmosphere interaction at different surface types in
# the mid- to lower reaches of Yangzi River Valley
**W.D. Guo[1], X.Q. Wang[1], J.N. Sun[1], A.J. Ding[1], and J. Zou[1]**
[1]Institute for Climate and Global Change Research, School of Atmospheric Sciences, Nanjing
University, Nanjing, China
*Correspondence to*: W.D. Guo (guowd@nju.edu.cn), or J.N. Sun (jnsun@nju.edu.cn)
**Abstract**
The mid- to lower reaches of Yangzi River Valley is located within the typical East Asia monsoon zone.
Rapid urbanization, industrialization, and development of agriculture have led to fast and complicated
land use and land cover changes in this region. To investigate land-atmosphere interaction in this region
where human activities and monsoon climate are highly interactive with each other, micro-
meteorological elements over four different surface types, i.e. urban surface represented by the
observational site at Communist Party School in Nanjing (hereafter DX), suburban surface represented
by the ground site at Xianling (XL), and grassland and farmland represented by field sites at Lishui
County (LS-grass and LS-crop), are analyzed and their differences are revealed. Impacts of different
surface parameters applied for different surface types on the radiation budget and surface-atmosphere
heat, water, and mass exchanges are investigated. Results indicate that (1) the largest differences in
daily average surface air temperature ($T_a$), surface skin temperature ($T_s$), and relative humidity (RH) ,
which are found during the dry periods between DX and LS-crop, can be up to $3.21^{o}C$, $7.26^{o}C$, and
22.79% respectively. During the growing season, the diurnal ranges of the above three elements are the
smallest at DX and the largest at LS-grass, XL and LS-crop; (2) differences in radiative fluxes are
mainly reflected in upward shortwave radiation (USR) that is related to surface albedo and upward
longwave radiation (ULR) that is related to $T_s$. USR is the smallest and ULR is the largest at DX.
During the growing season, the average difference in ULR between the DX site and other sites with



vegetation cover can be up to $20 Wm^{-2}$. The USR variability is the largest at LS-crop, while the diurnal
variation of ULR is the same as that of $T_s$ at all the four sites; (3) the differences in daily average
sensible heat (H) and latent heat (LE) between DX and LS-crop are larger than 45 and $95 Wm^{-2}$,
respectively. The proportion of latent heat flux in the net radiation ($LE/R_n$) keeps increasing with the
change of season from the spring to summer. XL site demonstrates a distinct forest feature; (4) surface
albedo is small while the Bowen ratio is large at DX (an urban site). The urban heat island effect results
in higher $T_a$ and $T_s$ at DX site that is $2^{o}C$ higher than that at other sites in the nighttime. It is found that
surface albedo and roughness length variability both increase at LS-crop during the harvest season and
straw burning periods. LE is dominant due to irrigation. Negative H is observed since evaporative
cooling leads to low $T_s$. Daily variability of $T_s$ and $T_a$ is the lowest at LS-crop while RH is the largest.
In the summer, the grassland albedo at XL site gradually becomes larger than that at the sites in Lishui.
Since the forest-like effects becomes more distinct at XL, $LE/R_n$ increases rapidly. Thereby, although $T_s$
is higher at XL than that at LS-grass , there is no large difference in $T_a$ between the two sites.

## 1 Introduction

Land use/Land cover change (LULCC) is one of the most important anthropogenic forces to weather
and climate change in local ,regional and global scale (IPCC, 2013). On earth, over 80% of the total
land surface has been affected by human activities (Sanderson,2002) in the form of construction and
farmland and loss of forest, and are increasing greatly at multiple spatial and temporal scales in regions
of different climate regimes.(Davin and De Noblet-Ducoudré, 2010; Kalnay and Cai, 2003; Lawrence et
al., 2012). Logging and creation of new farmlands have changed land use in the tropics (DeFries et al.,
2002); intensive human activities in temperate regions have changed forests and grasslands to farmlands,
while urbanization and industrialization has been intensifying all the time and desert area has been
expanding (Gao et al., 2003; Suh and Lee, 2004). In boreal regions, forests have degraded to grasslands
and farmlands due to fire and pests damages as well as logging (Brown et al., 2010; Lohila et al., 2010).
Under the same climate background, the radiation components and surface energy distribution are





controlled by characteristic surface factors such as vegetation cover, albedo, roughness length, etc.
(Amiro et al., 2006; Feddema et al., 2005; Jin and Roy, 2005), and subsequently affect micro-
meteorological elements of temperature, humidity, and precipitation. The effect of LULCC on regional
and global climate has been documented through the climate models. Large-scale vegetation
degradation and the development of agriculture and animal husbandry in different scale will lead to
decreases in precipitation (Mcalpine et al., 2009; Werth and Avissar, 2002), while LULCC will affect
the temperature difference between the surface and air temperature and vegetation feedback ,such as
tropical warming and boreal cooling due to deforestation and urban heat island (Arnfield, 2003;
Bounoua et al., 2002;Luyssaert et al,2014; Pielke et al., 2002).
However, a severe uncertainty in models still exists due to the insufficient knowledge of the surface-
atmosphere interaction in response to variations in surface fluxes and energy balance (Bonan, 2008;
Wang and Eleuterio, 2001; Pitman et al., 2009).One way to solve this problem is verify the model and
parameterization schemes by driving them with field measurement and observations, which is really
important in present study. With the development of a new tool, Fluxnet (Baldocchi, 2001), a large
number of land surface pair sites were built up from wild, rural to urban and produce many significant
results. Both management on existing types of land cover and conversion to a different type can affect
the local climate (Baldocchi, 2014). The areodynamically rougher and darker oak savanna has higher $R_n$,
H and $T_a$ than the grassland in the same climate condition (Baldocchi and Ma, 2013; Baldocchi et al,
2004); deforestation would have a cooling effect on $T_a$ in mid- to high latitudes and a warming effect in
low latitude (Lee et al., 2011); Wildfires on different land cover make different effects (Krishman et al.,
2012; Montes-Helu et al., 2009); the management practices of rangeland and cropland or the change of
crop types can influence the energy balance and water budget (Alberto et al., 2009; Alberto et al.,2011;
Baldocchi and Rao, 1995; Coulter et al., 2006; Masseroni et al., 2014). Besides, in a city, LE at
residential site is less dependent on short-term precipitation that at grass site and H is related with land
cover and building intensity (Offerle et al., 2006).
China, with the largest population in the world, is one of the fastest growing and urbanizing economies.
So LULCC has an significant influence on the regional to global climate change by altering the land



surface energy and water flux in China ( Zhao and Pitman, 2005; Suh and Lee, 2004; Chen et al., 2014).
Most field sites are built in the arid and semi-arid region. In the northeastern ecotone between
agriculture and animal husbandry, farmland has a greater roughness and energy fluxes than grassland in
Tongyu (Feng et al., 2012) but less than reed wetland in Panjin (Li et al., 2009). In the degraded
grassland in West China,  the oasis-desert transition zone is a cold source relative to the Gobi in
Dunhuang (Wang et al., 2005), and energy fluxes are different over different land surface due to
vegetation, precipitation and soil moisture in Loess Plateau (Wang et al., 2010). Besides, rapid urban
expansion has changed heat fluxes in the Pearl River delta a lot (Lin et al., 2009) and has increased
sensible heat flux in Beijing (Zhang et al., 2009). There are obvious differences between different
surface types, including air temperature, soil moisture and surface radiation and energy budget (Zhao et
al., 2013; Zhang et al., 2014). But in monsoon region, even though the changes in surface heat fluxes
can influence monsoon onset or weakening and  precipitation (Hsu and Liu, 2003; Fu and Yuan, 2001;
Qiu, 2013; Xue et al., 2004) ,both flux observations and studies are very limited (Bi et al., 2007; Lee et
al., 2011) ,especially over the mid- to lower reaches of Yangzi River Valley.
The mid- to lower reaches of Yangzi River Valley is located in the typical East Asian monsoon region,
where the land use and land cover has been experiencing rapid changes with more complicated land use
types due to the rapid urbanization, industrialization, and development of agriculture and animal
husbandry. Interaction between human activities and monsoon climate is most intensive in this region.
Under the background of monsoon climate, studies about the differences in the diurnal and seasonal
variations of the land-atmosphere interaction over various surface types are almost blank in this region.
In order to better understand the characteristics and mechanisms for the exchanges of mass, energy, and
water vapor between the land surface and atmosphere in the mid- to lower reaches of Yangzi River
Valley, in the present study we analyze observations collected at several ground sites over different
surface types around Nanjing. These sites include a school site in the urban area (hereafter DX), the
Xianling site in suburban Nanjing (XL), a grassland site (LS-grass) and a farmland site (LS-crop) in
Lishui County, which is located in the countryside. Data used in this study were collected at these sites
in the spring and summer of 2013. The goals of the study are (1) to compare the seasonal and diurnal



variation of micro-meteorological elements over different land surface type, (2) to reveal the differences
in surface radiation budget, energy distribution between various surface types; and (3) to calculate
important surface parameters over different surface and investigate the feedback of different surface
types to the atmosphere and its impact on local climate. The mechanisms for the surface-atmosphere
feedback will be further investigated. This study will fill the gap of observation scarcity in land-
atmosphere interaction in the mid- to lower reaches of Yangzi River Valley, and provide scientific
evidences for the regional climate simulation and climate change prediction.
**2 Data and methodology**
**2.1 Introduction of field sites**
The observations used in this study were collected at four field sites located in urban, suburban, and
countryside areas of Nanjing. The four sites are referred to as DX, XL, LS-grass and LS-crop hereafter.
The DX site (Fig. 1a) is located at Baixia District of Nanjing (32°2′24″N,118°47′24″E), which is the
central urban area of Nanjing. Residential and commercial buildings are dominant within the 500m
radius centered around the DX site, and thereby the land surface type is a typical urban surface at this
site. Average height of buildings is 19.7m, and the building coverage is up to 70%.
The XL Site (Fig. 1b) is the key station in the experiment of Station for Observing Regional Processes
of the Earth System, Nanjing University (SORPES-NJU). It is located at (32°7'13" N, 118°57'9" E, 43m
above sea level) in the eastern suburb of Nanjing, an upwind area along the prevailing wind direction in
Nanjing. The distance between the XL Site and DX Site is 18km. Within the 50m×50m area at the XL
site, the grass height is 7cm. Outside the site area are woodlands from afforestation with a height of
around 3m. The XL site is located inside the Xianling campus of Nanjing University. Since its operation
in 2011, continuous observations are measured through a suite of equipment instruments. The
observations include conventional meteorological measurements at various levels, surface energy
budget measurements, boundary layer meteorological elements measurements, surface radiation
measurements, atmosphere components and aerosols measurements, etc. The data used in this study are
standard measurements at half-hour intervals.
The site (31°43′08″N,118°58′51″E ) at Lishui county is taken as a satellite site of the SORPES-NJU.
The distance between Lishui site and DX site is 38km. Lishui site consists of a pair of observational
sites, one over the grassland (LS-grass, Fig. 1c) and the other (LS-crop, Fig. 1d) over the farmland
nearby. The grass height is about 60cm at the LS-grass, and the observation period is from January 2012
to February 2014. Rice grows at LS-crop in the summer (mid June to early November) and winter
wheat grows in the winter (from mid- to late November to early June of next year). The maximum
height of wheat is 75cm. The observation period at LS-crop is from January 2013 to February 2014. The
distance between the two sites at Lishui is 1.62km.
**2.2  Micro-meteorological measurements**
The instruments used at the XL site include automatic weather station, eddy covariance system (EC),
energy balance system, and soil temperature / humidity observation system. Table 1 lists the major
measured variables, ranges, observation heights, and instrument models. The same measurement
method is applied at the XL, LS-grass and LS-crop sites. In the DX site, there is no measurement of soil
moisture and soil temperature. The instrument of eddy-covariance and energy balance system are
installed at the top of 36.5m high tower on the roof of the building which is 22m high. The air
temperature and humidity can be observed by the  tower-mounted system 9m high above the roof.
The auto weather station (AG1000, Campbell) measures micro-meteorological elements of temperature,
pressure, relative humidity, wind speed and direction, precipitation, and surface radiation components
of upward/downward shortwave and longwave radiation fluxes. Ts is measured by infrared detection
sensor (IRTS-P, Apogee).
Momentum, sensible and latent heat fluxes are measured by the eddy covariance system (EC3000,
Campbell), which includes a three-dimensional sonic anemometer (CSAT-3) and a infrared analyzer
(LI7500) at 3m height. The sampling frequency is 10Hz for measurements by the Data acquisition
(CR5000). Strict correction and quality control have been performed for all the turbulence



measurements. Coordinate rotation correction (Wilczak et al., 2001), frequency response correction
(Moore, 1986), and WPL correction etc. are applied in this study.
The soil heat flux plate (HFP01SC-L, Hukseflux) is at the depth of 8cm. At LS-grass and LS-crop sites,
the soil heat flux is measured at 5cm and 10cm below the ground, respectively. Soil temperature and
moisture at 5cm, 10cm, 20cm, 40cm, and 80cm are measured using Soil Temperature Profile Sensor
(STP01-L, Hukseflux) and Water Content Analyzer (S616-L, Cambell). No soil temperature and
moisture measurements are conducted at the DX site.
The data collected at the spring and summer (from March to August) of 2013 are used in the present
study. This is because the measurements are relatively complete during this period, which is also the
time when land-atmosphere interaction is strong.
**2.3  Methodology**
**2.3.1  Distribution of surface energy**
In the surface with fractional vegetation cover, surface energy budget can be expressed as
$$R_n = H + L_E + G_0 + R_e \qquad (1)$$
Where $R_n$ is the net radiation, H and LE are the sensible and latent heat fluxes respectively, $G_0$ is the
soil heat flux at the surface, $R_e$ is the remaining term, which is associated with the photosynthesis and
respiration of plants as well as vegetation and soil thermal storage, etc. (Burba et al., 1999; Harazono et
al., 1998). While in the urban areas, the energy balance must take anthropogenic and net storage heat
flux but not $G_0$ into consideration (Oke, 1987). In this paper, we only discuss the relationship between
H, LE and $R_n$ on the basis of the observation.
$R_n$ can be calculated from the four radiation components. Sensible and latent heat fluxes are calculated
by the following equations:
$$H = \overline{\rho} c_p \overline{w'T'} \qquad (2)$$
$$L_E = \overline{\rho} L_V \overline{w'q'} \qquad (3)$$



where $\rho$, $C_p$ and L are the air density(kg m$^{-3}$), the specific heat capacity at constant pressure(J kg$^{-}$
$^1$ K$^{-1}$), and latent heat of vaporization(J kg$^{-1}$). w', T' and q' are perturbations of vertical velocity
(m/s), temperature (K), and mixing ratio of water vapor (g/kg), respectively. Strict quality control has
been conducted for all the flux measurements.
**2.3.2  Parameters related to the land surface process**
Surface albedo can be calculated based on the equation below (Zhang et al., 2004):
$$\alpha = \frac{\sum R_{su}}{\sum R_{sd}}$$   (4)
where $R_{su}$ is the surface reflected radiation at half-hour interval, $R_{sd}$ is the solr radiation that reaches the
surface. This method to a certain degree can avoid the adverse influence of low albedo on the
calculation of daily average solar radiation when the solar zenith angle is too low. Daily average albedo
is the ratio between the upward and downward solar radiation at half-hour interval during the period
from 6:00 to 18:00 LST.
Following the same approach used in Li (2015), Bowen ratio is calculated based on the ratio of H and
LE. It is expressed as:
$$\beta = \frac{\sum H}{\sum L_E}$$   (5)
H and $L_E$ are sensible and latent heat fluxes at half-hour interval, respectively. Daily Bowen ratio is the
ratio between the sum of sensible and latent heat fluxes at half-hour interval over the entire day. The
ratio between sensible and latent heat fluxes at the same time is taken as the Bowen ratio at that same
time.
Following the independent method proposed by Chen (1993), which determines z0m using only the
mean wind speed and turbulence measured by ultrasonic anemometer, we fit the non-dimensional wind
speed ku / u * to the stability parameter z / L in a double logarithmic coordinate and obtain the value of



$ku/u_*$ under neutral condition. It is then applied to wind profile equation under neutral condition and
yields:
$$z_{0m} = (z-d)e^{-\frac{ku}{u_*}}$$    (6)
where u is the horizontal wind speed (m/s); k is the Von Karman constant, which is set to be 0.4 in this
study (Prueger et al., 2004); z is the height of the instrument probe (m); d is the zero displacement,
which is 2m at XL site, 0.4m at LS-grass site and 0.5m at LS-crop site. $u^*$ is the friction velocity (m s⁻
¹); $z_{0m}$ is the aerodynamic roughness length. Liu (2015) verified this independent method using
measurements from the semi-arid region in China.
**3  Results and discussion**
**3.1  Differences in micro-meteorological elements**
The year 2013 is a typical hot and dry year in China, especially in the mid- to lower reaches of Yangzi
River Valley (Wang et al., 2015). Fig. 2 shows the daily variations of air temperature, surface
temperature, and relative humidity at the four field sites. Surface temperature is calculated based on
measured upward and downward longwave radiation and the Stefan-Boltzmann law. Realistic daily
changing trends of temperature and humidity are displayed for the four sites, and maximum values of
air temperature and surface temperature both occur in August. The changing trend of relative humidity
is similar to that of precipitation, and the relative humidity tends to reach saturate at stations where there
is more precipitation and higher temperature. Fig.2 shows clearly that large differences in temperature
and humidity between the four sites mainly appear at April and August, when precipitation is relatively
small. The largest air temperature difference of 3.21°C is found between DX and LS-crop sites at the
beginning of August, and the largest surface temperature difference is 7.26°C. The largest relative
humidity difference is 22.79%. Apparently, even in the same climate background, there exist significant
differences in micro-meteorological elements between various surface types. Such differences are more
distinct when there is no precipitation or precipitation is relatively small. Generally, surface temperature



increases when vegetation cover fraction decreases except in the farmland, which is affected by
irrigation. This is consistent with findings from some experiments in mid-latitudes of North America
(Lee X. et al., 2011; Li et al., 2015).
Table 2 clearly indicates that, except that the minimum summer temperature is found at XL site,
extremely high/low values of seasonal average air temperature, surface temperature, and relative
humidity all occur at either LS-crop site or DX site, which are the two sites that are most affected by
human activities.
Fig. 3a and 3b suggest that the diurnal variations of both air temperature and surface temperature exhibit
single-peak feature in the spring and summer. The minimum value occurs at 7:00 LST in the morning
and the maximum value occurs in the afternoon. The air temperature variation lags that of the surface
temperature, and it also lags in the summer that in the spring. Due to the influence of the surface,
diurnal surface temperature range is larger than the diurnal range of air temperature. Except for the LS-
crop site, surface temperature is higher than air temperature in all the other three sites. Nighttime air
temperature and surface temperature at the DX site is higher than that in other sites by nearly $2^{\circ}$C due to
the urban heat island effects. Comparing the land surfaces that have vegetation cover, the grass height is
low at the XL site and the peak surface temperature variation is large with the largest temperature up to
$37.61^{\circ}$C. The peak surface temperature remains low at the LS-crop site due to irrigation, and even lower
than the daily maximum air temperature in the summer. The peak surface temperature at the LS-crop
site is only $32.4^{\circ}$C. Comparing the diurnal temperature ranges at the four sites, it is found that the
diurnal air temperature and surface temperature ranges are $4.79^{\circ}$C and $9.26^{\circ}$C in the spring, respectively,
which are relatively small. In the summer, the LS-crop site is covered by water due to irrigation and the
diurnal surface temperature range is only $7.77^{\circ}$C, which is the minimum among all the four sites. The
diurnal air temperature range is $6.86^{\circ}$C at LS-grass site in the summer, and the range is relatively large
among all the sites. Meanwhile, the diurnal surface temperature range at XL site is $12.46^{\circ}$C in the
summer, the largest among the four sites. Despite the large diurnal surface temperature range at XL site,
air temperature and diurnal air temperature range are not that large. This is because the afforest



woodlands surrounding the XL site promotes the heat flux exchange between the surface and
atmosphere.
Fig. 3c shows that the relative humidity is always larger in the summer than in the spring. Daily
maximum relative humidity occurs at around 7:00 am in the morning, and the minimum value occurs at
16:00 in the afternoon. The occurrence of the maximum an minimum values in the spring lags that in
the summer. The maximum value is found at LS-crop, with the summer average maximum value of
90.34%. The smallest relative humidity is found at the DX site, where the maximum summer average
humidity is only 58.72%. The diurnal relative humidity range at the four sites is larger in the spring than
in the summer, and the largest value is found at LS-crop site in the spring and at LS-grass in the summer
with the value of 39.03% and 27.36% respectively. The diurnal relative humidity range is the smallest at
the DX site, which is 23.29% in the spring and 20.40% in the summer. Fig. 3 clearly indicates that
different surface types can lead to differences in surface temperature and impose significant impacts on
micro-meteorological elements such as air temperature and relative humidity (Krishnan et al., 2012;
Luyssaert et al., 2014).
**3.2 Surface net radiation and energy distribution**
**3.2.1 Distribution of net radiation**
Fig. 4 displays the daily variation of the four components of surface radiation flux, i.e. the downward
shortwave radiation (DSR), upward shortwave radiation (USR), downward longwave radiation (DLR),
and upward longwave radiation (ULR). DSR and USR are mainly affected by clouds and aerosols in the
atmosphere. In the monsoon region of the mid- to lower reaches of Yangzi River Valley, the cloudy and
rainy weather is dominant during the period of May to July, leading to lower shortwave radiation
despite the higher solar zenith angle. Under the same climate background, DSR and DLR are similar at
the four sites. However, large differences are found in USR and ULR at the four sites. This is because
USR is related to surface albedo while ULR is associated with surface temperature. Daily maximum
values of USR and ULR both occur in early August. The maximum value of USR are 48.67、55.29、




35.80 and 52.19Wm$^{-2}$ at the LS-grass, LS-crop, DX, and XL sites respectively. The maximum value of
ULR at the sour sites are 515.22、492.78、529.59 and 518.81W m$^{-2}$ respectively.
USR changes following the changes in DSR and surface albedo. Variability of monthly average USR
(Fig.5b) is similar to that of DSR (Fig. 5a), and both are the smallest at the DX site. However, compared
to that in other sites, the USR at the LS-crop decreases rapidly since May and reaches its minimum of
14.87 Wm-2 at the end of June. This is because of the albedo decrease at the LS-crop site, which is
caused by straw burning at the end of May after the winter wheat harvest. Rice starts growing since late
June, and the USR at LS-crop site becomes similar to that at other sites by August. The ULR remains
largest at the DX site and smallest at LS-crop site, which is attributed to the increases in vegetation
cover fraction from May to August and irrigation at the LS-crop site. The difference in ULR between
the DX and LS-crop can be up to 26.9 W.m-2 in August.
With the same weather and climate background, there are no significant differences in DSR and DLR
among the four sites, despite their distinct seasonal differences. The maximum daily DSR are around
550 W m$^{-2}$ and 600W m$^{-2}$ in the spring and summer respectively, and the maximum daily DLR are
about 370 W m$^{-2}$ and 450 W m$^{-2}$ in the spring and summer respectively. Fig. 6c shows that the
maximum daily average USR at the LS-crop site is smaller in the summer than in the spring by 16.98
Wm-2, which is different from the situation in the other three sites, where surface albedo increases in
the summer due to the decreased vegetation cover fraction. As a result, the USR decreases by 90.35、
84.79、59.49 W m$^{-2}$ at the LS-crop, XL, and DX sites respectively. The diurnal variation of ULR (Fig.
6d) depends on diurnal variation of surface temperature (Fig.2b). The largest ULR occurs at XL site in
the daytime and at DX site in the nighttime. The maximum ULR and diurnal ULR range both are the
smallest at the LS-crop site due to irrigation.
**3.2.2  Surface energy distribution**
The land-atmosphere energy exchange is the driving force for the local climate and is under great
influence of climate change (Reale and Dirmeyer, 2000;Li et al., 2009). Fig. 7 shows daily variation of
net radiation ($R_n$), sensible heat flux (H), and latent heat flux (LE). $R_n$ and DSR have the  similar



changing trends and both are small during the monsoon precipitation period. The average $R_n$ during the
growing season is different over different surface types, and the values are 126.55, 118.40, 112.58,
105.08W m$^{-2}$ at the LS-crop, LS-grass, XL, and DX sites respectively. The average value of H during
the growing season are 4.62, 39.99, 26.13, 53.48 W m$^{-2}$ respectively at the four sites, while LE are
74.11, 53.59, 59.73, and 34.45 W m$^{-2}$ respectively. The above results suggest that under the same large
scale forcing, there exist distinct differences in radiation and turbulent fluxes over different surface
types. Such kinds of differences are the largest between LS-crop and DX sites, where human activities
are the most intensive among the four sites. During the non-precipitation period, differences in $R_n$ and H
are large in July and August, with absolute value of differences up to 79.88 and 166.56 W m$^{-2}$
respectively. At the beginning of April, with little precipitation and insufficient soil moisture content,
irrigation at the LS-crop site leads to the LE difference to be up to 107.87 Wm-2. The above differences
are largely caused by the difference in vegetation cover at the surface and associated with the growth of
vegetation and accompanied water cost.
Monthly average $R_n$ reaches the largest in July and the value at LS-crop is 170.37 Wm$^{-2}$. Since the
rainy season starts, the proportion of LE in $R_n$ gradually increases. Although the monthly variation of $R_n$
are similar at the four sites (Fig. 8a), there exist large differences in sensible and latent heat flux (Fig. 8b,
8c). H is smallest at the LS-crop. Rice planting starts in mid June and the surface is covered by water.
Negative sensible heat flux occurs in July and August at the LS-crop site. The difference in sensible
heat flux between the LS-crop and DX sites is 44.86W m$^{-2}$. The change of LE is opposite to that of H.
LE reaches the largest in July and August with the value greater than 95 Wm$^{-2}$. LE is 55.68 W m$^{-2}$
larger at the LS-site than at the DX site.
Fig. 9 depicts the seasonal average surface energy components. H accounts for a large proportion of $R_n$
in the spring, with the value ascending from 25.60%,47.65%,60.92% to 75.45% at LS-crop, LS-
grass, XL, and DX sites. In the summer, accompanied with the rainy season, vegetation thrives and the
ratio of LE/$R_n$ significantly increases. The values are 60.01%, 47.18%, 66.65% and 37.86% respectively
at the four sites. Again effects of the woodlands surrounding the XL site are reflected in the



measurements at XL. The negative sensible heat flux is attributed to the negative difference in air
temperature and surface temperature (Table 1) at the LS-crop site.
Fig. 10 shows the diurnal variations of net radiation, sensible heat flux, and latent heat flux for the
spring and summer at the four sites. $R_n$ is negative at the nighttime and the maximum value occurs at
around 14:00 in the daytime. The differences in peak value of $R_n$ between the spring and summer are
larger than 50 W m$^{-2}$ at all the four sites. Except for the DX site, the difference in maximum H between
the spring and summer is greater than 30 W m$^{-2}$ at the other three sites. The difference in the peak value
of LE between the spring and summer is larger than 60 W m$^{-2}$. At the DX site, H is always larger than
LE in both the spring and summer. Fig. 10b shows clearly that the peak value of H is largest at the XL
site in the spring, and at DX site in the summer. The differences between these two sites and the LS-
crop site are 92.53 and 162.21 W m$^{-2}$ respectively. The peak value of H is the smallest at the LS-crop
site, and H can be negative during the entire day in the summer. This is because the LS-crop site is
covered by water in the summer, and the large evaporation results in low surface temperature that is
lower than air temperature (Lee et al., 2004). For both the spring and summer, the peak value of LE
remains largest at the LS-crop site and smallest at the DX site. The difference in LE between the two
sites can be up to 138.46 W m$^{-2}$ in the spring and 156.46 W m$^{-2}$ in the summer. This result suggests that
there exist distinct differences in radiation and surface energy fluxes over different underlying surface
types not only in the semiarid region (Wang et al, 2010; Li et al., 2015), but also in the monsoon region
of mid- to lower reaches of Yangzi River valley.
**3.3  Mechanism analysis**
Changes in the surface characteristics are always accompanied by variations in the parameters involved
in land surface process. Differences in characteristic parameters such as albedo, Bowen ratio, roughness
length, etc affect radiation and energy distribution, which subsequently feedback to the atmosphere and
affect micro-meteorology in the surface layer (Amiro et al., 2006;Lee et al., 2011).
**3.3.1  Radiation and turbulent exchange coefficients**



In land surface processes, albedo is a basic parameter that affects net radiation in the surface (Li et al.,
2015; Krishnan et al., 2012; Zhang et al., 2014). Daily variations of albedo at the four sites (Fig.11a)
show that albedo decreases with the growing of vegetation, and the rapid decrease is found in the LS-
crop site with the largest daily decrease of 0.13. At the beginning of June, albedo decreases to less than
0.09 due to the straw burning and later remains less than 0.1 due to irrigation. Since mid July, the
albedo at the LS-crop site gradually increases to 0.15 accompanied with the growing of rice, and
becomes close to that at DX and XL sites. At DX site, there is no large daily variation of albedo, which
remains at around 0.13. The difference in albedo determines the daily USR variation at the four sites
(Fig. 4b). Fig. 12 shows that except for XL site, the albedo at the other three sites is smaller in the
summer than in the spring, which is mainly related to the growing of vegetation. At the XL site,
possibly because of insufficient precipitation after mid July, the summer albedo increases and becomes
slightly larger than that in the spring. If not considering irritation in the LS-crop site, albedo always
decreases with the increase of vegetation cover fraction, while the radiative forcing leads to increases in
surface temperature. In the boreal region, however, model studies and field experiments both have
revealed that (Defries et al., 2002; Lee et al., 2011) degradation of vegetation represented by
deforestation could lead to lower surface temperature. This is quite different from the situation in
temperate and tropical regions. Thereby, the warming and cooling trends might be different over
regions with different surface types and background climate, which makes it important to conduct
mechanism study for the impact of different land cover types on local temperature.
Bowen ratio is the measure of surface energy distribution. It reflects the dry and wet condition of the
surface to a certain degree (Li et al., 2015; Wang et al., 2010). The daily variation of the Bowen ratio
(Fig. 11b) indicates that large variation occurs in the spring and there exist distinct differences between
the four sites. The difference between the DX and LS-crop sites is larger than 10 at the beginning of
March. The differences between the four sites and the variation at each site both decrease during the
rainy season. Except for the DX site, Bowen ratio is smaller than 1.0 since early May at all the other
three sites and LE becomes dominant. Considering the surface types at the four sites, it is found that
daily variation of Bowen ratio is small (stable) at the surface with large vegetation cover fraction and



high soil moisture content, which is more capable of adjusting the heat and water balance. This result is
consistent to that for the semiarid region (Hu et al., 2009). Fig. 12b suggests that with more
precipitation and large vegetation cover fraction in the summer, the Bowen ratio is much smaller in the
summer than in the spring. Comparing the Bowen ratio at the four sites, the largest value is found at the
DX site while the smallest is at the LS-crop site for both the spring and summer. The negative sensible
heat flux at the LS-crop site in the summer makes the Bowen ratio to be less than zero. At the XL site,
H/LE further decreases due to the effects of woodlands nearby, while $LE/R_n$ further increases and
accounts for a larger proportion in the energy distribution (Fig. 9b).
Generally speaking, LE accounts for a large proportion of $R_n$ at sites where Bowen ratio is small. Since
the relative humidity is affected by temperature and water vapor content, it will increase with the
decrease in Bowen ration and temperature (Li et al., 2015). This relation is basically satisfied in the
mid- to lower reaches of Yangzi River Valley, but the general situation becomes more complicated due
to the influences of many other factors.
**3.3.2 Surface roughness length at different surface types**
Surface roughness length is an important ecological and land surface parameter. The spring-summer
average roughness length at the DX site calculated based on the shape of the surface roughness
elements is 2.82m, which has no distinct seasonal variation. Monthly variations of surface roughness
length at the other three sites are shown in Fig. 13, which shows that the roughness length basically
increases with the month from May to August. The differences in roughness length between the four
sites are largely caused by the differences in vegetation cover, which are rice, grass, and lawn at the LS-
crop, LS-grass, and XL sites respectively. The roughness lengths at the three sites during the growing
season are 0.05m, 0.02m, and 0.17m respectively. Apparently the roughness length at the XL site more
reflects the characteristics of the woodlands nearby. The roughness length decreases slightly in July at
XL and LS-grass sites due to insufficient precipitation. In early June after the harvest of winter wheat,
the roughness length at the LS-crop is less than 0.01m, but it gradually increases later with the growth
of rice.





Fig. 14 shows that from the spring to summer, the increase in roughness length at the XL site is much
larger than that at the LS-grass site, and the differences between the spring and summer at the two sites
are 0.045m and 0.007m respectively. This is attributed to the differences in vegetation type and height.
This result indicates that different land use type and roughness element height are responsible for the
different roughness length at various time scales.
Comparing results at LS-grass and XL sites, both are covered by natural vegetation, it can be found that
the vegetation with larger roughness length can promote stronger turbulent flux transfer and has a
higher capability of temperature adjustment. This explains why average temperature and diurnal
temperature range both are relatively small at XL site, despite the high surface temperature at this site
(Fig. 3a). Sensitivity experiments of a numerical model study also demonstrated that the surface
roughness length is one of the most sensitive factors for land-atmosphere exchange (Liu et al., 2015).
**4  Conclusions and discussion**
2013 is a typical dry and hot year. During the growing season in the mid- to lower reaches of Yangzi
River Valley monsoon region, the four different surface types, i.e. urban surface, woodland, grassland,
and cropland, can directly affect the surface radiation balance and land-atmosphere exchanges of heat,
water vapor, and mass fluxes, and subsequently affect local climate. In the present study, we have
revealed the differences in several physical parameters between the four typical surface types mentioned
above during the study period and explored the mechanisms for the differences.
Daily variations of the micro-meteorological elements at the four sites are different due to different
surface characteristics. The differences in micro-meteorological elements are more distinct during the
dry and hot period. The differences between the DX and LS-crop sites are the most significant. The
largest differences in air temperature, surface temperature, and relative humidity between the two sites
are 3.21$^{\circ}$C, 7.26$^{\circ}$C, and 22.79% respectively. Compared with that over the land use type covered by
natural vegetation (LS-grass and XL sites), albedo at the urban surface is smaller and thus the radiative
forcing is stronger, leading to higher surface temperature. However, insufficient moisture content makes



the Bowen ratio large. Hence the surface heat is transferred to the atmosphere mainly in the form of
sensible heat flux, and air temperature is high while relative humidity is small. Meanwhile, the urban
heat island effect results in higher surface $T_a$ and $T_s$ in the nighttime at the DX site that is 2$^{\circ}$C higher
than that at other sites, and the diurnal temperature range is small. At Lishui county, the crops were not
stressed by lack of moisture or high temperatures during the growing period due to irrigation, so latent
heat flux dominates the land-atmosphere heat flux exchange. Surface temperature and air temperature
both are relatively low while the relative humidity is relatively large due to large evaporation at the
surface. Surface albedo reaches its smallest value in June because of wheat harvest and straw burning at
this time. Daily variation of USR increases under the influence of albedo. For both spring and summer,
the peak value of diurnal variation of surface temperature and diurnal temperature range are the smallest
at the DS-crop site, mainly because the sufficient soil moisture content at this site acts to lower the
surface temperature. Negative sensible heat flux is found at this site in the summer due to the large
evaporation. Compared with the situation over surface types with natural vegetation cover, peak value
in the diurnal variation of surface temperature and its diurnal range both are large at the XL site, where
vegetation cover fraction is low. However, the woodland nearby the XL promotes turbulent exchange
and heat flux transfer, leading to lower air temperature and its diurnal range. From the spring to summer,
latent heat flux becomes dominant with the increase of albedo, and the Bowen ratio gradually decreases
to less than 1. Diurnal ranges of Air temperature, surface temperature, and relative humidity all
gradually decrease.
Under the same climate background, changes in surface albedo result in changes in the radiative forcing.
The Bowen ratio change caused by the surface energy distribution and the aerodynamic resistance
change related to surface roughness length jointly determine the differences in surface temperature, air
temperature, and relative humidity between different land surface types with various vegetation cover.
The monsoon precipitation and land use changes by human activities makes the land-atmosphere
interaction more complicated. Compared with the situation at sites with natural vegetation cover, air
temperature at the XL site is smaller than that at the LS-grass site, whereas the surface temperature is
higher than that at the LS-grass site. Such a inconsistency is caused by the complexity in the surface



characteristics. The present study has investigated the features and mechanisms of land-atmosphere
interaction over four different surface types. However, contributions of various land surface parameters
to micro-meteorological elements are different, and further quantitative analysis of the contribution of
each individual parameter is necessary.
**Acknowledgements**
This research is jointly sponsored by Natural Science Foundation of China (Grant No. 41475063,
91544231), the National Science and Technology Support Program (2014BAC22B04), and Program for
New Century Excellent Talents in University. This work is also supported by the Jiangsu Collaborative
Innovation Center for Climate Change.





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



1    **Table 1.** Instruments, measurement ranges, measurement heights, and instrument models.

| Parameter/Variable Name Description | Range | Measurement Height | Instrument |
|---|---|---|---|
| Wind speed sensor | 0–45 m/s | 2.0m | Met One, 014A-L |
| Humidity probe | 0–100% | 2.0m | Vaisala,HMP45C-L |
| Temperature probe | -45–60 ℃ | 2.0m | Vaisala, HMP45C-L |
| Barometric pressure sensor | 600–1060 millibar | 8.0m | Vaisala, CS105 |
| Tipping bucket rain gage | 0–15 mm | 0.3m | TE525MM-L, R.M Young |
| Pyranometer (SW flux) | 0–1200 W m$^{-2}$ | 1.5m | Kipp & Zonen, CM21 |
| Pyrgeometer (LW flux) | 0–700 W m$^{-2}$ | 1.5m | Kipp & Zonen, CG4 |
| 3-D Sonic anemometer | | 3.0m | Campbell, CSAT-3 |
| Opened path infrared $CO_2/H_2O$ analyzer | | 3.0m | Li-Cor, LI7500 |
| Water content reflectometer | 0–70 VV$^{-1}$ | 5,10,20,40,80 | CAMPELL, CS616-L |
| Soil temperature profile | -50–70 ℃ | 2,5,10,20,50,80 | Hukseflux, STP01-L |
| Soil heat flux plate | -300–300 W m$^{-2}$ | 8 cm | Hukseflux, HFP01SC-L |



**Table 2.** Seasonal averages of Air temperature, surface temperature, and relative humidity at the four
sites.

| Site name | LS-grass | | LS-crop | | DX | | XL | |
|---|---|---|---|---|---|---|---|---|
|  | MAM | JJA | MAM | JJA | MAM | JJA | MAM | JJA |
| Ta(℃) | 17.29 | 29.85 | 16.44 | 29.02 | 17.50 | 29.92 | 16.53 | 28.64 |
| Ts(℃) | 16.72 | 29.11 | 16.02 | 28.02 | 18.76 | 31.23 | 17.98 | 30.12 |
| RH(%) | 69.51 | 76.60 | 71.41 | 78.53 | 60.88 | 68.61 | 63.44 | 73.54 |



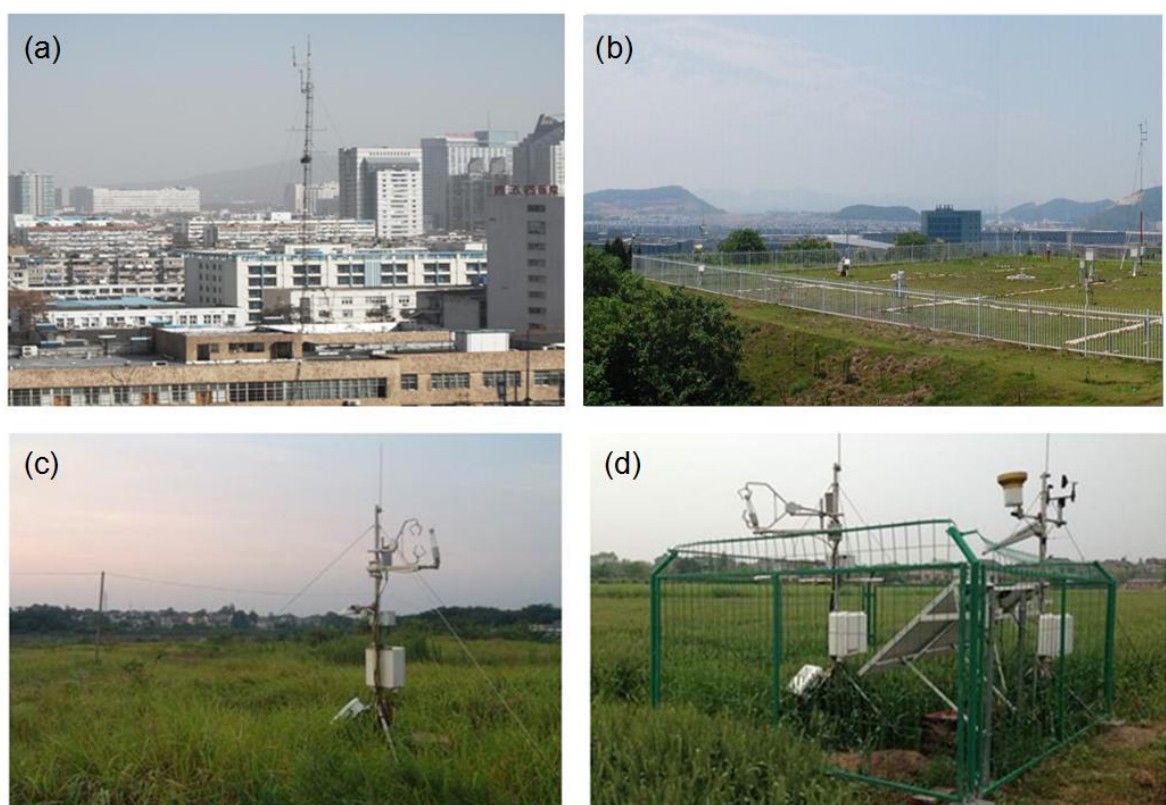

Figure 1. The four field sites at (a) DX, (b) XL, (c) LS-Grass and (d) LS-Crop in Nanjing.





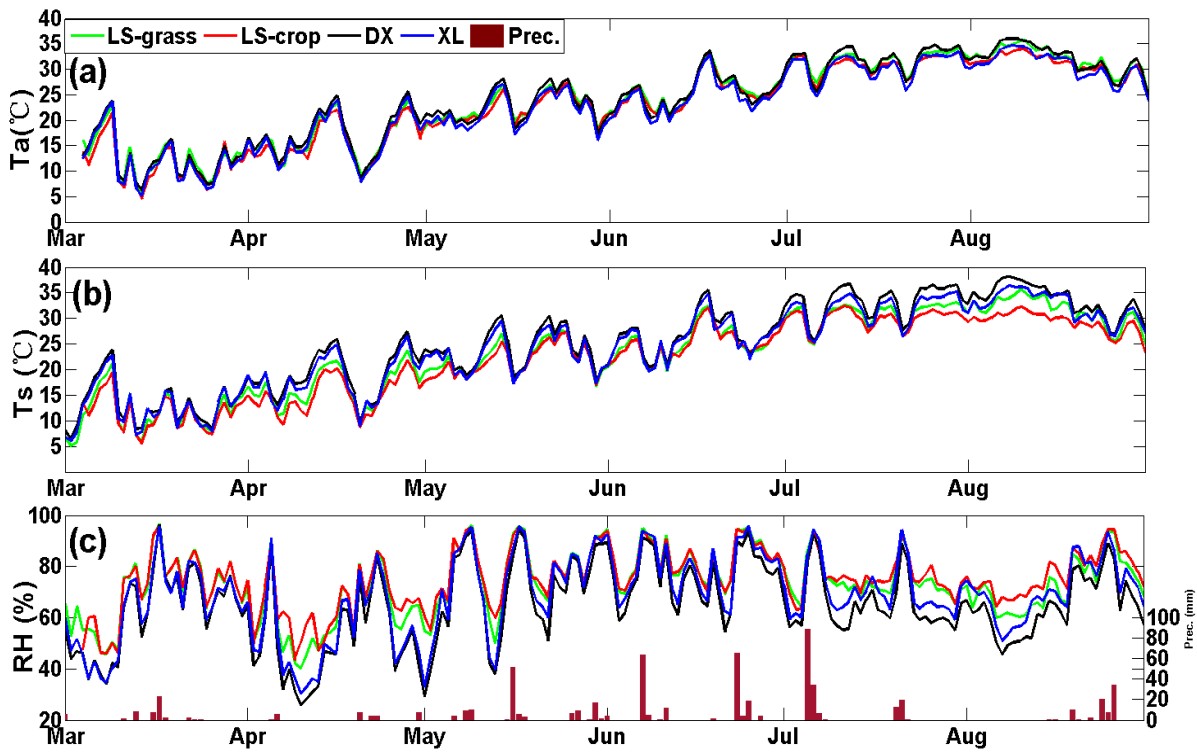

**Figure 2.** Daily variations of (a) air temperature, (b) surface temperature, and (c) relative humidity at the four sites in Nanjing from March to August, 2013.



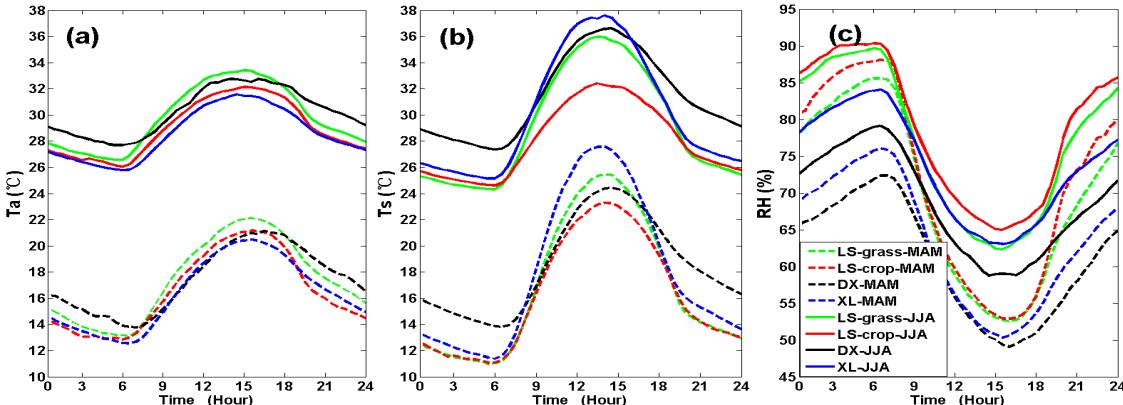

**Figure 3.** Diurnal variations of (a) air temperature, (b) surface temperature, and (c) relative humidity at
the four sites in Nanjing in the spring and summer.





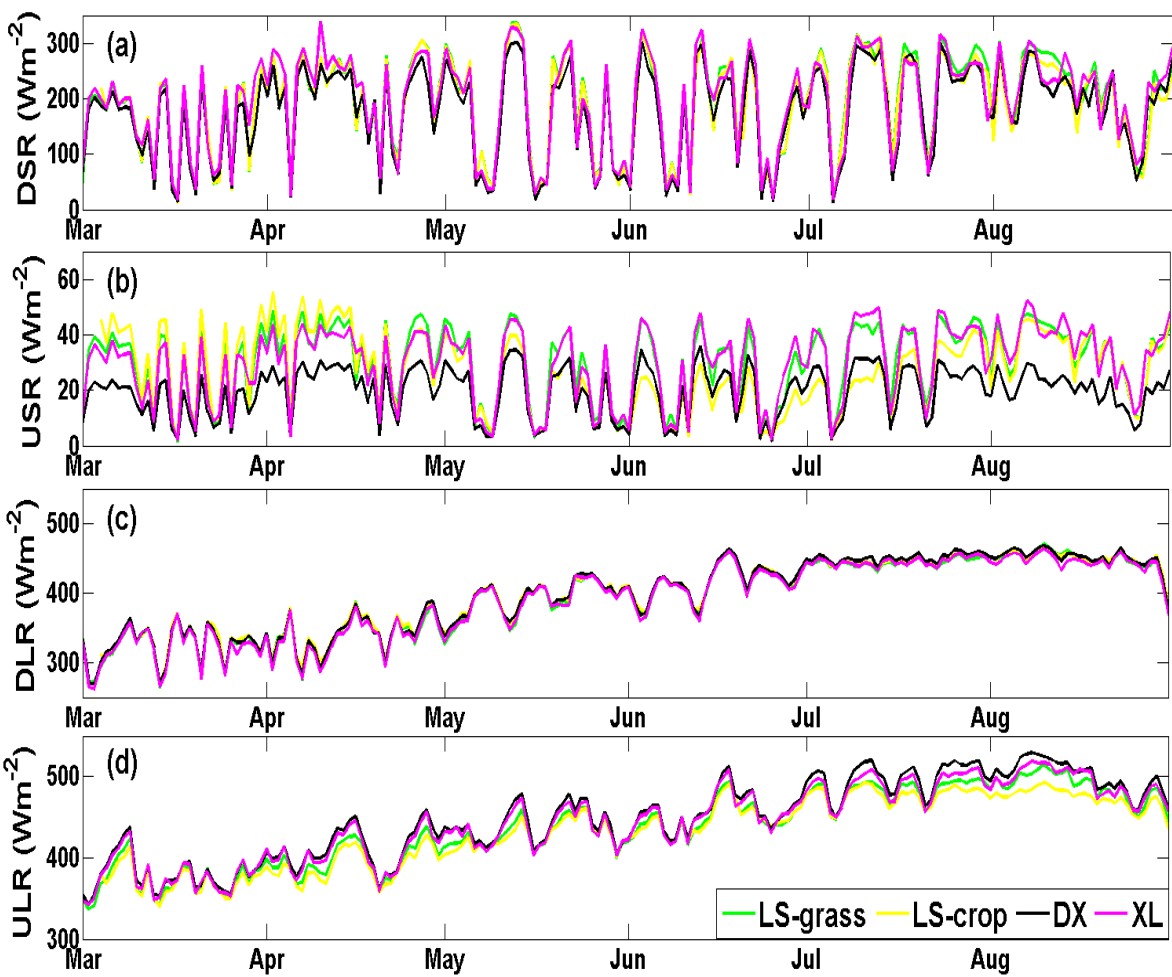

**Figure 4.** Daily variation of (a) downward shortwave radiation (DSR), (b) upward shortwave radiation (USR), (c) downward longwave radiation (DLR), and (d) upward longwave radiation (ULR) at the four sites in Nanjing.



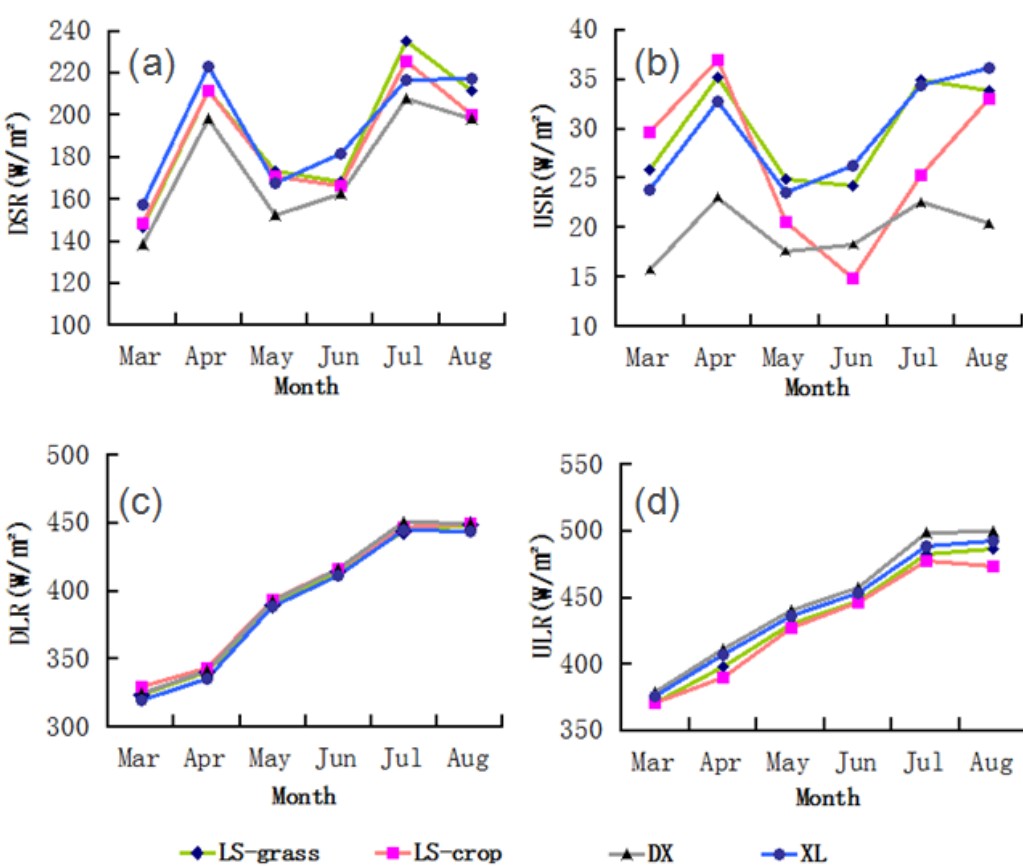

**Figure 5.** Monthly variation (a) downward shortwave radiation (DSR), (b) upward shortwave radiation (USR), (c) downward longwave radiation (DLR), and (d) upward longwave radiation (ULR) at the four sites in Nanjing.



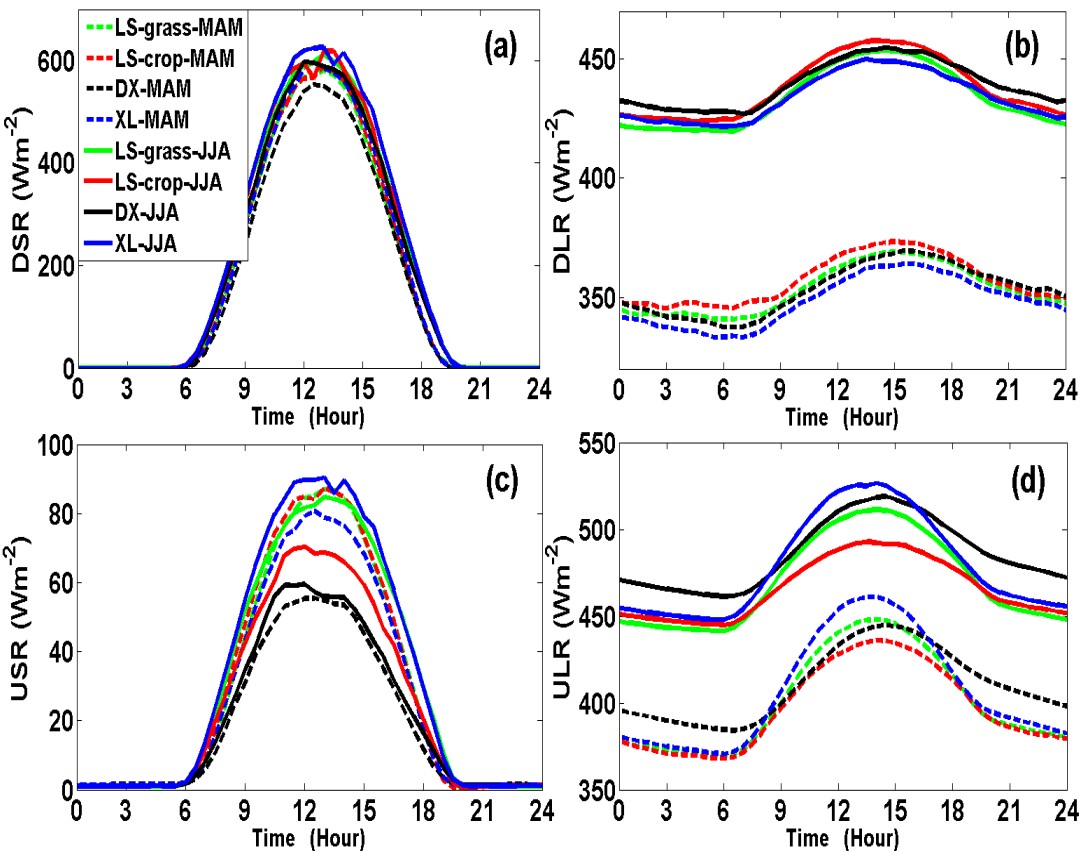

**Figure 6.** Diurnal variation of (a) downward shortwave radiation (DSR), (b) downward longwave radiation (DLR), (c) upward shortwave radiation (USR), and (d) upward longwave radiation (ULR) at the four sites in Nanjing.





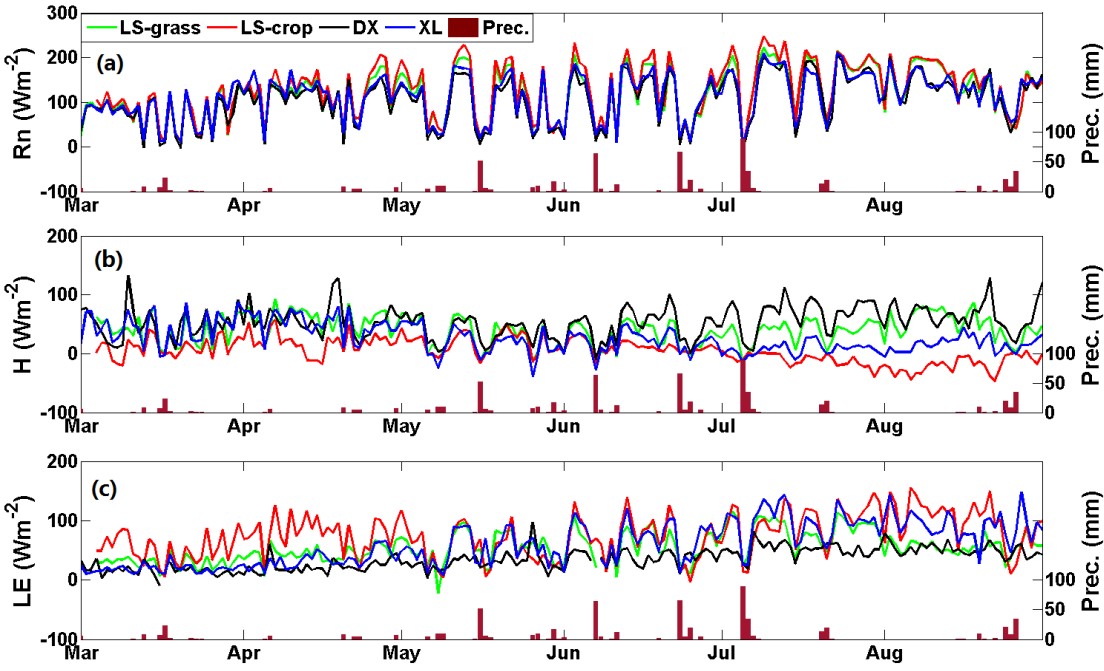

3  **Figure 7.** Daily variations of (a) net radiation, (b) sensible heat flux, (c) latent heat flux at the four sites

4  in Nanjing from March to August, 2013.





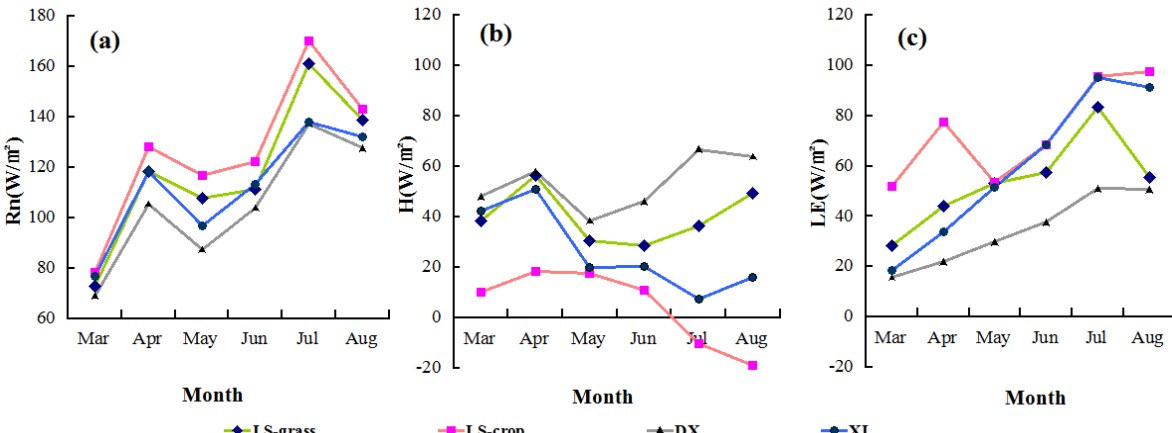

3  **Figure 8.** Monthly variation of (a) net radiation, (b) sensible heat flux, (c) latent heat flux at the four

4  sites in Nanjing from March to August, 2013.

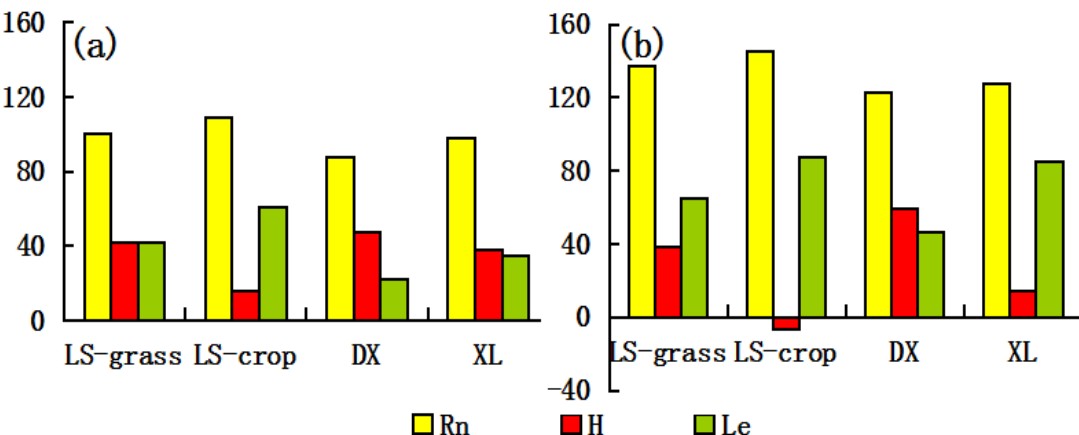

3  **Figure 9.** Seasonal average distribution of surface energy for the (a) spring and (b) summer at the four

4  sites in Nanjing.



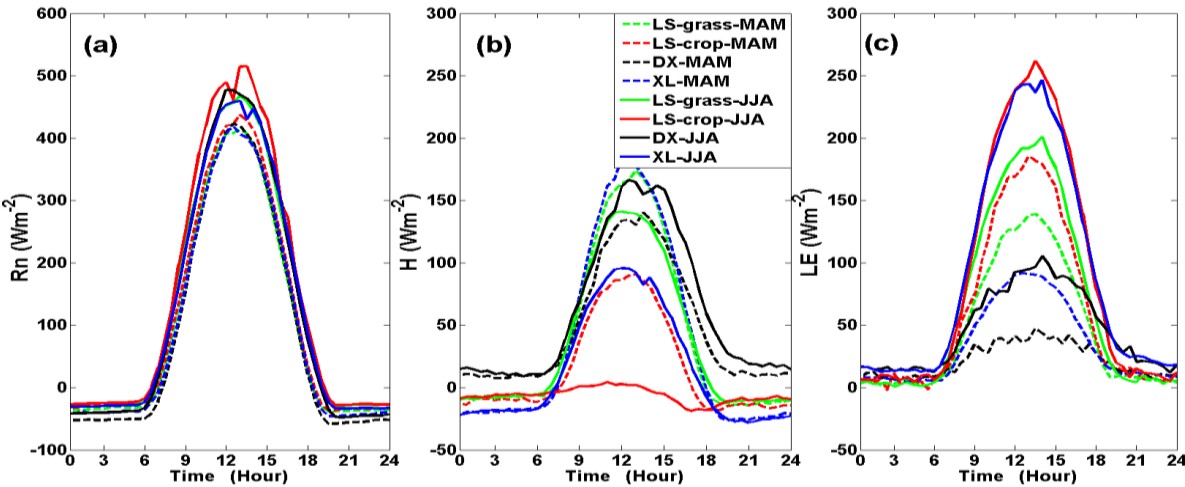

**Figure 10.** Diurnal variation of (a) net radiation, (b) sensible heat flux, (c) latent heat flux at the four

sites in Nanjing from March to August, 2013.



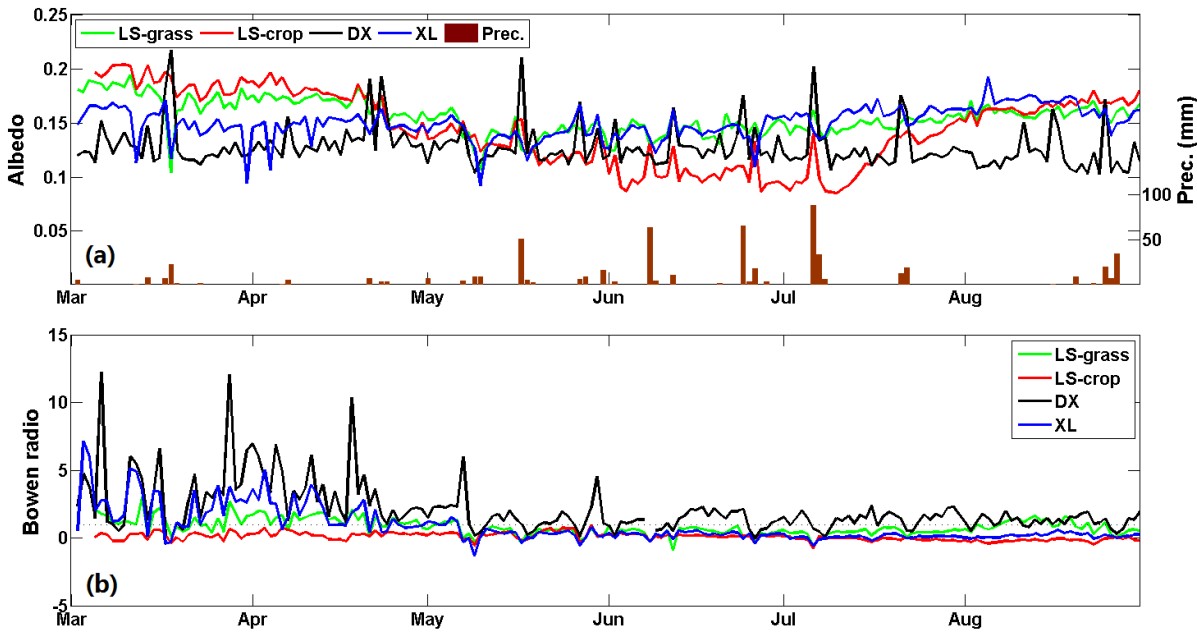

3  **Figure 11.** Daily variation of (a) albedo and (b) Bowen ratio at the four sites in Nanjing from March to

4  August, 2013.



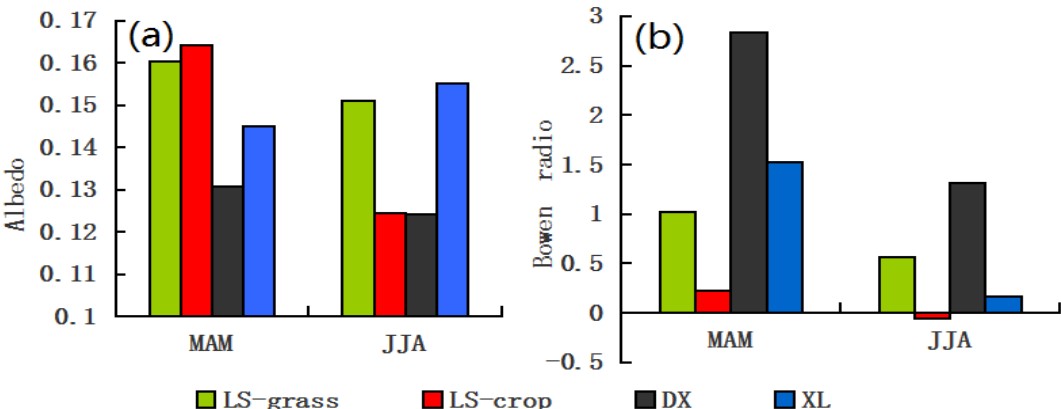

3    **Figure 12.** Seasonal averages of (a) albedo and (b) Bowen ratio for the spring and summer at the four

4    sites in Nanjing.





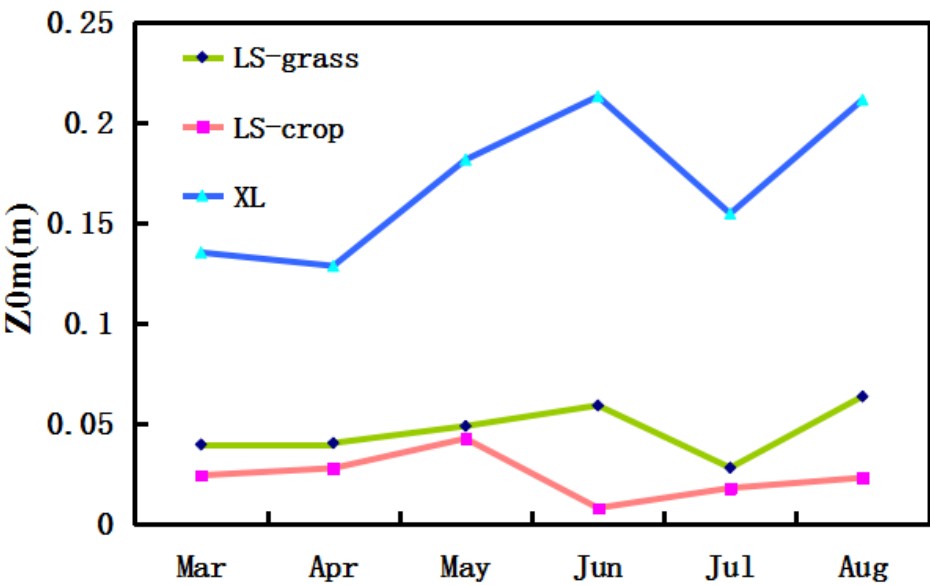

**Figure 13.** Monthly variations of surface roughness length at the three sites in Nanjing from March to
August, 2013.





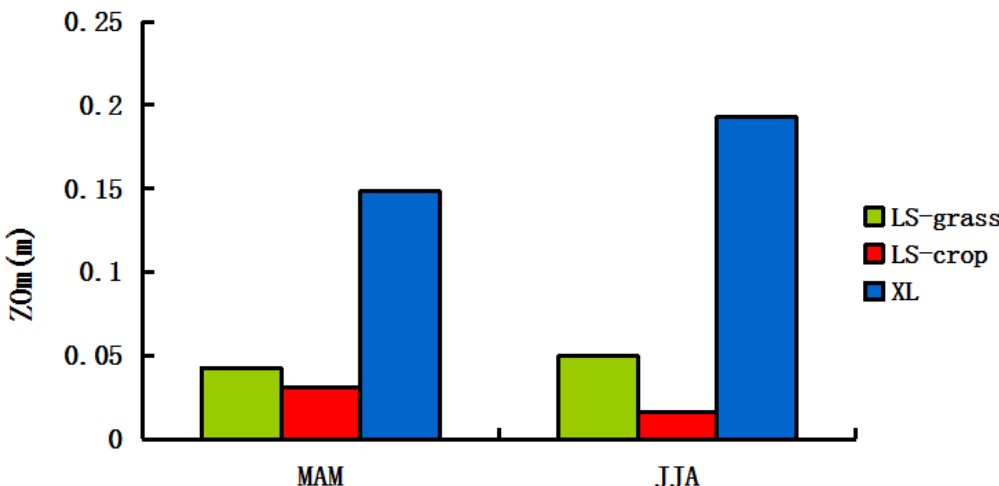

3 **Figure 14.** Seasonal averages of surface roughness length at the four sites in Nanjing for the spring and

4 summer of 2013.