# Peer review of "Comparison of land-atmosphere interaction at different surface types in the mid- to lower reaches of Yangtze River Valley"

_Atmospheric Chemistry and Physics, 2016_

## Referee Comment (RC1) · Anonymous Referee #1 · 23 Feb 2016

With the development of social economy and population density increases, evidences have indicated that land use and environmental quality change a lot at a global scale, and the surface ecosystem becomes increasingly fragile. The surface vegetation cover has serious deteriorated by multi-sources (including NOAA-AVHRR and TIROS-TOVS satellite remote sensing data), which has been largely documented. While land cover changes, land variables such as albedo, roughness, and bulk transfer coefficients, also change, which lead to the variation of surface heat fluxes, and then result in surface temperature anomaly. Therefore, in many previous studies, using dramatic land condition change to evaluate the land surface process impact has been widely used for regional land surface impact studies in preliminary stage to excite more comprehen-

sive studies. And most of them focused on the importance of land surface processes through climate modeling. This research investigated the impacts of different surface parameters for four different surface types over the mid-to lower reaches of Yangzi River on the radiation budget and surface-atmosphere water, heat and mass exchanges. Firstly, the authors revealed the differences in several physical parameters among the four typical surface types. Secondly, they tried to explore the mechanism for the differences. The analyses in the paper are well organized and the results are reasonable. Few published papers discuss the differences of surface physical parameters among different surface types based on the field observations, especially over the mid- to lower reaches of Yangzi River Valley. This paper provides useful information, especially for the land-atmospheric interaction research over East Asia monsoon region. The presentation of this article is generally clear. I suggest publication of this paper with some revisions.

1. In the first part of the introduction, the authors review many studies about the impacts of land cover change on global and regional climate. Land-atmosphere interaction is strong in East Asian monsoon zone. Since the research focus on Yangzi River, some previous work on LULCC effects on China or East Asian climate should be mentioned importantly in the introduction. Actually, there were serious land degradations over East Asia during the past several decades and have identified Tibet Plateau, Northwest China and Inner Mogonial were among areas with severe land degradation. For example, Xue et al. (1996) and Qian and Xue (2010) have pointed out the East Asia summer monsoon circulation was weakened and the precipitation is reduced due to the land degradation over three areas. 2. In Table 1, the units of the measurement height for soil temperature and water content are not specified. It should be cm? Please complete them. 3. In page 10, line 10 and 11, "......, and it also lags in the summer that in the spring" please clarify the sentence. It's so hard for readers to understand. Also, in the same line, "Due to the influence of the surface,.......", it's a general statement and in the research article, it should be avoided. It's better to state what the influences are and how the surface or other anthropogenic factors make the surface temperature

show larger diurnal range than air temperature. 4. In the results and discussion part, the authors show the differences of many observed elements and several surface characteristics over four sites during spring and summer. Actually, tables that can show the detailed quantitative differences could be the compliment for the figures (e.g. figure 2, figure 3. . ...). For example, for the figure 3, a table can be presented that show the average of diurnal air temperature, surface temperature and relative humidity for the four sites. The table and figure can exhibit the differences more easily observable.

---

## Short Comment (SC1) · 14 Mar 2016

General comments:

This manuscript revealed the differences of land-atmosphere interactions in four typical land cover types (Urban surface, Suburban surface, Grassland surface and Cropland surface). It is well organized and written. I suggest this manuscript for publication in Atmospheric Chemistry and Physics after some minor revisions and corrections.

1. DX-Urban and XL-Suburb terms in the manuscript are suggested to replace corresponding DX and XL terms, and then land cover type will be distinguished more easily like LS-crop and LS-grass terms.

2. Nighttime surface/air temperature differences are mainly emphasized in the manuscript, but daytime surface/air temperature differences are rarely discussed. From Figure 3, we can see that daytime urban site surface/air temperature is lower than suburb site or grass site, it is just the opposite of the existing results based on remote sensing LST data and meteorological station data, extra explanations or discussions about the contrary daytime surface/air temperature results are needed in this paper. Following existing publications are recommended for reference:

Liu, S., Jiang, R., Wang C. Wang Y.: Observation analysis on spatial and temporal distribution characteristics of summer urban heat island in Nanjing (in Chinese), Trans Atmos Sci, 37(1): 19-27, 2014

Zeng, Y., Qiu, X. F., Gu, L. H., He, Y. J., Wang, K. F.: The urban heat island in Nanjing, Quaternary International, 208(1), 38-43, 2009.

Zhou, D., Zhao, S., Liu, S., Zhang, L., Zhu, C.: Surface urban heat island in China's 32 major cities: Spatial patterns and drivers, Remote Sensing of Environment, 152, 51-61, 2014.

3. Similar descriptions or explanations such as "albedo decrease with the growing of vegetation" are mentioned in the manuscript many times, for example: Page15, line2-3, line9-10 and line12-14, etc. It is not appropriate for this paper in my opinion. Firstly, it can be seen that albedo increase with growing of the paddy rice from Figure 11, and this fact is mentioned in Page15, line5-7. Secondly, relations between albedo and vegetation fraction is not fixed, albedo may increase with the growing of vegetation according to existed studies. Therefore, descriptions or explanations such as "albedo decrease with the growing of vegetation" should be used with caution in order to avoid misleading the readers. Please refer the following papers:

Gao, F.: MODIS bidirectional reflectance distribution function and albedo Climate Modeling Grid products and the variability of albedo for major global vegetation types. Journal of Geophysical Research, 110, D01104, 2005.

Rechid, D., Raddatz, T.J., Jacob, D.: Parameterization of snow-free land surface albedo as a function of vegetation phenology based on MODIS data and applied in climate modelling. Theoretical and Applied Climatology, 95, 245-255, 2009.

Wang, K., Liang, S., Schaaf, C. L., Strahler, A. H.: Evaluation of Moderate Resolution Imaging Spectroradiometer land surface visible and shortwave albedo products at FLUXNET sites. Journal of Geophysical Research, 115, D17107, 2010.

Specific comments:

1. Page11, line19: USR should be DLR, USR is affected by albedo, not clouds and aerosols in the atmosphere.

2. Page12, line15-18: It is hard to understand USR at the LS-crop site is smaller in the summer than in the spring as a result of albedo increase(line17-18), I think it is a mistake. The phenomenon that albedo at the LS-crop site in summer is smaller than that in spring can be seen clearly in Figure 12. The sentence "where surface albedo increases in the summer due to the decreased vegetation cover fraction" is also hard to understand, because vegetation cover fraction is supposed to increase with the paddy rice growing in summer, please explain this sentence more.

3. Page12, line18-19: The meaning of the sentence "As a result, the USR decreases by 90.35ãĂĄ84.79ãĂĄ59.49 W m-2 at the LS-crop, XL, and DX sites respectively." is apparently not corresponding to the Figure 6c, and LS-crop should be LS-grass because LS-crop is analyzed before. At LS-grass, XL and DX sites, USR all increases? Please confirm it.

4. Page16, line19-22: roughness lengths are not right according to Figure 13 and Figure 14, please change "0.05m, 0.02m,and 0.17m" to "0.02m, 0.05m,and 0.17m".

---

## Referee Comment (RC2) · Anonymous Referee #2 · 6 May 2016

The comparison studies for data analysis from paired observational sites under same (or similar) climate background could reveal the differences of energy budgets which resulted by land surface characteristics directly and quantitatively. The mid- to lower reaches of Yangzi Rivers is located within the East Asia Monsoon zone, and the mechanism of LULCC is complicated because of the interaction between the general circulation and human activities. The four surface types selected in this study are the most typical in the region. The paper is well organized and written, I suggest it will be published after some revision.

1. A subplot is suggested to be added in Fig 1, which content the location of 4 sites with satellite background. It will be better understanding than written-description. 2. I also

suggest the DX and XL are replaced by DX_urban and XL_suburb; 3. In P12, L1-2, this sentence should be present in part 2.3.1, after the variables description. Is there any more QA/QC consideration for eddy covariance data processes? 4. The approximate irrigation schedule should introduce in the part of LS_crop site description; 5. In Fig 11. There exist obvious high correlation between albedo and precipitation for LS_crop and DX sites and low correlation between LS_grass and XL sites, I suggest the authors give some interpretation. 6. Page 16, L9-10, the variation for RH is mainly affected by synoptic system, it is hard to depict it varies with the Bowen ration and temperature.
* * *

---

## Author Comment (AC1) · 13 Jun 2016

**Response to Referee #1**

*With the development of social economy and population density increases, evidences have indicated that land use and environmental quality change a lot at a global scale, and the surface ecosystem becomes increasingly fragile. The surface vegetation cover has serious deteriorated by multi-sources (including NOAA-AVHRR and TIROS-TOVS satellite remote sensing data), which has been largely documented. While land cover changes, land variables such as albedo, roughness, and bulk transfer coefficients, also change, which lead to the variation of surface heat fluxes, and then result in surface temperature anomaly. Therefore, in many previous studies, using dramatic land condition change to evaluate the land surface process impact has been widely used for regional land surface impact studies in preliminary stage to excite more comprehensive studies. And most of them focused on the importance of land surface processes through climate modeling. This research investigated the impacts of different surface parameters for four different surface types over the mid-to-lower reaches of Yangtze River on the radiation budget and surface-atmosphere water, heat and mass exchanges. Firstly, the authors revealed the differences in several physical parameters among the four typical surface types. Secondly, they tried to explore the mechanism for the differences. The analyses in the paper are well organized and the results are reasonable. Few published papers discuss the differences of surface physical parameters among different surface types based on the field observations, especially over the mid-to-lower reaches of Yangtze River Valley. This paper provides useful information, especially for the land-atmospheric interaction research over East Asia monsoon region. The presentation of this article is generally clear. I suggest publication of this paper with some revisions.*

**Response:** We would like to thank the referee for providing the insightful suggestions, which indeed help us reconsider and further explore the underlying problems in comparing the land-atmosphere interaction at different surface types in the mid-to-lower Yangtze River valley. In the revised manuscript, we have added more descriptions on the research background and in-depth discussion of the differences in micro-climate elements and mechanism analysis.

*Major comments:*

*• In the first part of the introduction, the authors review many studies about the impacts of land cover change on global and regional climate. Land-atmosphere interaction is strong in East Asian monsoon zone. Since the research focus on Yangtze River, some previous work on LULCC effects on China or East Asian climate should be mentioned importantly in the introduction. Actually, there were serious land degradations over East Asia during the past several decades and have identified Tibet*

*Plateau, Northwest China and Inner Mogonial were among areas with severe land degradation. For example, Xue et al. (1996) and Qian and Xue (2010) have pointed out the East Asia summer monsoon circulation was weakened and the precipitation is reduced due to the land degradation over three areas.*

**Response:** Accepted. The references of previous work on LULCC effects on East Asia has been added in the introduction in the revised manuscript (P4, lin14-17).

• *In Table 1, the units of the measurement height for soil temperature and water content are not specified. It should be cm? Please complete them.*

**Response:** We have added the units of the measurement height for soil temperature and water content, it's "cm" in Table 1.

• *In page 10, line 10 and 11, "......., and it also lags in the summer that in the spring" please clarify the sentence. It's so hard for readers to understand. Also, in the same line, "Due to the influence of the surface,........", it's a general statement and in the research article, it should be avoided. It's better to state what the influences are and how the surface or other anthropogenic factors make the surface temperature show larger diurnal range than air temperature.*

**Response:** Thanks. We rephrased the sentence "*and it also lags in the summer that in the spring* " as " The peak time of both air and surface temperature in spring lags that in summer." in P10, line 11-13. Besides, the difference of radiation budget on land surface between daytime and nighttime results in larger diurnal range of surface temperature than air temperature. The words "*Due to the influence of the surface*" was unclear for readers to understand, so we rewrote it in P10, line 28-29.

• *In the results and discussion part, the authors show the differences of many observed elements and several surface characteristics over four sites during spring and summer. Actually, tables that can show the detailed quantitative differences could be the compliment for the figures (e.g. figure 2, figure 3.....). For example, for the figure 3, a table can be presented that show the average of diurnal air temperature, surface temperature and relative humidity for the four sites. The table and figure can exhibit the differences more easily observable.*

**Response:** Accepted. Figure 2 has already shown the average of micro-meteorological elements for the four sites in different season, and we added the diurnal range of these elements below figure 2 in the revised manuscript.

---

## Author Comment (AC2) · 13 Jun 2016

**Response to Referee #2**

*The comparison studies for data analysis from paired observational sites under same (or similar) climate background could reveal the differences of energy budgets which resulted by land surface characteristics directly and quantitatively. The mid- to lower reaches of Yangtze Rivers is located within the East Asia Monsoon zone, and the mechanism of LULCC is complicated because of the interaction between the general circulation and human activities. The four surface types selected in this study are the most typical in the region. The paper is well organized and written, I suggest it will be published after some revision.*

**Response:** We would like to appreciate the referee for providing the insightful suggestions, which indeed help us reconsider and further explore the the differences of land-atmosphere interaction at different surface types in the mid-to-lower Yangtze River valley. In the revised manuscript, we have added more clear descriptions on the location of the pair sites and comparison on physical characteristics with different land cover, as well as in-depth discussion concerning the mechanism.

*Major comments:*

*• A subplot is suggested to be added in Fig1, which content the location of 4 sites with satellite background. It will be better understanding than written-description.*

**Response:** Thanks. We have added the subplot in Figure 1. It will be easier for readers to know the location and surface types.

*• I also suggest the DX and XL are replaced by DX_urban and XL_suburb.*

**Response:** Accepted. "DX" and "XL" have been replaced by "DX-urban" and "XL-suburb" in the revised manuscript.

*• In P12, L1-2, this sentence should be present in part 2.3.1, after the variables description. Is there any more QA/QC consideration for eddy covariance data processes?*

**Response:** We rechecked the sentence in P12, line1-2, and there may be some misunderstandings. QA/QC is definitely a crucial issue for the proper use of eddy covariance data. In section 2.2, the QA/QC is mentioned as follows: "Strict correction and quality control (Foken er al., 2004) have been performed for all the turbulence measurements. Coordinate rotation correction (Wilczak et al., 2001), frequency

response correction (Moore, 1986), and WPL correction etc. are applied in this study."

• *The approximate irrigation schedule should introduce in the part of LS_crop site description;*

**Response:** We have added the schedule of agricultural activities in the part of LS-crop site description in the part of LS-crop site description in P6, line 11-13.

• *In Fig 11. There exist obvious high correlation between albedo and precipitation for LS_crop and DX sites and low correlation between LS_grass and XL sites, I suggest the authors give some interpretation.*

**Response:** It is human activities that results in the high correlation between albedo and precipitation for LS-crop and DX-urban sites but not for LS-grass or XL-suburb sites. At urban site, roof of the building is nearly watertight, the waterlogging after raining leads to a high albedo in a short time. In cropland, the soil with sparse vegetation cover has high soil wetness during the growing season. When being covered by water after rainfall event, the albedo increases immediately. This phenomenon has been explained in the part of 3.3.1.

• *Page 16, L9-10, the variation for RH is mainly affected by synoptic system, it is hard to depict it varies with the Bowen ration and temperature.*

**Response:** Accepted. We rewrote this sentence in the revised manuscript. The variation of RH is not attributed only to vertical turbulent exchange, but also advection. Temperature and water vapor can not fully explain the change of RH in P16, line 18-23.

---

## Author Comment (AC3) · 13 Jun 2016

**Response to Short Comment from Scientific Community #1**

*This manuscript revealed the differences of land-atmosphere interactions in four typical land cover types (Urban surface, Suburban surface, Grassland surface and Cropland surface). It is well organized and written. I suggest this manuscript for publication in Atmospheric Chemistry and Physics after some minor revisions and corrections.*

**Response:** We would like to thank the referee for providing the insightful comments, which indeed help us reconsider and further explore the underlying problems when we analyze the difference of land-atmosphere interaction at different surface types in the mid-to-lower Yangtze River valley. In the revised manuscript, we have added more clear descriptions on the physical characteristics of climate elements and surface parameter, as well as the discussion of mechanism.

*General comments:*

*• DX-Urban and XL-Suburb terms in the manuscript are suggested to replace corresponding DX and XL terms, and then land cover type will be distinguished more easily like LS-crop and LS-grass terms.*

**Response:** Accepted. These replacements help the readers more easily to understand.

*• Nighttime surface/air temperature differences are mainly emphasized in the manuscript, but day time surface/air temperature differences are rarely discussed. From Figure 3, we can see that daytime urban site surface/air temperature is lower than suburb site or grass site, it is just the opposite of the existing results based on remote sensing LST data and meteorological station data, extra explanations or discussions about the contrary daytime surface/air temperature results are needed in this paper. Following existing publications are recommended for reference:*

*Liu, S., Jiang, R., Wang C. Wang Y.: Observation analysis on spatial and temporal distribution characteristics of summer urban heat island in Nanjing (in Chinese), Trans Atmos Sci, 37(1): 19-27, 2014*

*Zeng, Y., Qiu, X. F., Gu, L. H., He, Y. J., Wang, K. F.: The urban heat island in Nanjing, Quaternary International, 208(1), 38-43, 2009.*

*Zhou, D., Zhao, S., Liu, S., Zhang, L., Zhu, C.: Surface urban heat island in China＇s*

*32 major cities: Spatial patterns and drivers, Remote Sensing of Environment, 152, 51-61, 2014.*

**Response:** Firstly, as the first paper mentioned, UHI is evident in the nighttime but not typical in the daytime. Secondly, when discussing the intensity of UHI, we must take the climate background into consideration. As shown in the figure below, summer in 2013 is an extremely drought period in southern China, the precipitation decreased by more than 78% of the average amount, breaking the historical record over the past 50 years (Yuan et al., 2016), especially in the mid-to-lower reaches of Yangtze River (Hou et al, 2014; Zhao et al., 2015). We therefore have an assumption to explain the "contradictory" phenomenon mentioned above. In the urban area and cropland, human watering and irrigation or other activities alleviate the natural drought effect in these areas. But in the grassland and suburb area, lacking water limited evaporation cooling to a large extent. So the extreme drought induced higher temperature in the natural vegetation cover in 2013 than before but didn't have large influence in the area with intense human activities, and therefore not only weakened UHI but also made daytime urban site surface/air temperature lower than suburb site or grass site.

Reference

Hou W, Chen Y, Li Y, et al. Climatic characteristics over China in 2013 [J]. *Meteorological Monthly*, 2014, 40(4):482-493 (in Chinese).

Yuan W, Cai W, Yang C, et al. Severe summer heatwave and drought strongly reduced carbon uptake in Southern China [J]. *Scientific Reports*, 2016, 6(25):87–90.

Zhao J, Yang J, Gong Z, et al. Analysis of and Discussion about Dynamic-Statistical Climate Prediction for Summer Rainfall of 2013 in China[J]. *Advances in Meteorological Science & Technology*, 2015 (in Chinese).

[Figure]

Figure 1. Regional anomalies of air temperature (°C) (a), precipitation (mm) (b) and relative humidity (%) (c) during July-August 2013. All data compare 2013 and the average of 1960–2012. The provinces with bold black boundary lines are the study area in this study. The right-bottom figures show the boundary of South China Sea. The maps were created by the ArcMap 9.3. (Yuan et al., 2016)

• *Similar descriptions or explanations such as* "*albedo decrease with the growing of vegetation*" *are mentioned in the manuscript many times, for example: Page15, line23, line9-10 and line12-14, etc. It is not appropriate for this paper in my opinion. Firstly, it can be seen that albedo increase with growing of the paddy rice from Figure 11, and this fact is mentioned in Page15, line5-7. Secondly, relations between albedo and vegetation fraction is not fixed, albedo may increase with the growing of*

*vegetation according to existed studies. Therefore, descriptions or explanations such as "albedo decrease with the growing of vegetation" should be used with caution in order to avoid misleading the readers. Please refer the following papers:*

*Gao, F.: MODIS bidirectional reflectance distribution function and albedo Climate ModelingGridproductsandthevariabilityofalbedoformajorglobalvegetationtypes. Journal of Geophysical Research, 110, D01104, 2005.*

*Rechid, D., Raddatz, T.J., Jacob, D.: Parameterization of snow-free land surface albedo as a function of vegetation phenology based on MODIS data and applied in climate modelling. Theoretical and Applied Climatology, 95, 245-255, 2009.*

*Wang, K., Liang, S., Schaaf, C. L., Strahler, A. H.: Evaluation of Moderate Resolution Imaging Spectroradiometer land surface visible and shortwave albedo products at FLUXNET sites. Journal of Geophysical Research, 115, D17107, 2010.*

**Response:** Thanks. The words *"albedo decrease with the growing of vegetation" and "albedo always decreases with the increase of vegetation cover fraction"* is easy to mislead the readers. We have rewritten it as " Fig. 12 shows that except for XL-suburb site, the albedo at the other three sites decrease from spring to summer. At the XL-suburb site with sparse and low grass, possibly because of insufficient precipitation after mid-July, the summer albedo increases and becomes slightly larger than that in the spring. But at grassland, the albedo decreases largely in the green-up phrase, which results in the lower albedo in summer. And the dramatic decrease of surface albedo in early June is associated with the biomass burning due to the cultivation system in this region, i.e., a rotation of wheat in winter and rice in summer." in part of 3.3.1.

*Specific comments:*

*• Page11, line19: USR should be DLR, USR is affected by albedo, not clouds and aerosols in the atmosphere.*

**Response:** It has been corrected in P11, line 26.

*• Page12, line15-18: It is hard to understand USR at the LS-crop site is smaller in the summer than in the spring as a result of albedo increase(line17-18), I think it is a mistake. The phenomenon that albedo at the LS-crop site in summer is smaller than that in spring can be seen clearly in Figure 12. The sentence "where surface albedo increases in the summer due to the decreased vegetation cover fraction" is also hard to understand, because vegetation cover fraction is supposed to increase with the paddy rice growing in summer, please explain this sentence more.*

**Response:** We corrected it in the part of 3.2.1. Yes, it is easy to misunderstand the sentence that "*the maximum daily average USR at the LS-crop site is smaller in the summer than in the spring by 16.98 Wm-2 , where surface albedo increases in the*

*summer due to the decreased vegetation cover fraction*". Consider the seasonal variation, the decrease of albedo (Figure 12a) result in the decrease of USR (Figure 6c) at crop site from spring to summer. When it comes to the daily variation, albedo increases from June to August in P12, line 20-26.

*• Page12, line18-19: The meaning of the sentence "As a result, the USR decreases by 90.35、84.79、59.49 W m-2 at the LS-crop, XL, and DX sites respectively." is apparently not corresponding to the Figure 6c, and LS-crop should be LS-grass because LS-crop is analyzed before. At LS-grass, XL and DX sites, USR all increases? Please confirm it.*

**Response:** Corrected. We have rewrote this sentence as "USR at XL-suburb, LS-grass and DX-urban grows to 90.35、84.79、59.49 W m$^{-2}$ respectively in the summer." in P12, line 19-20 in our revised manuscript.

*• Page16, line19-22: roughness lengths are not right according to Figure 13 and Figure 14, please change "0.05m, 0.02m,and 0.17m" to "0.02m, 0.05m,and 0.17m".*

**Response:** Accepted. It has been changed.

---

## Author Response (AR2)

Cover Letter

Dear editor,

Thanks a lot for your excellent editorial work and consideration of the constructive comments from the reviewer. We have revised the manuscript according to your suggestions and the version is as follows. We hope that our replies and the revised manuscript are acceptable, and look forward to hearing from you with a decision regarding our paper. Thank you very much!

Sincerely yours,

Weidong Guo

Response to Editor

• *There are some mistakes in reference list, Please check and correct them.*

Corrected. The references of previous work related to the topic of the manuscript have been included in the sections of introduction and references in this revised manuscript.

• *Some references miss the doi number. Please try to add them.*

Done. The missed doi numbers have been added in the part of reference.

[revised manuscript text omitted]